# Food Price Volatility and Asymmetries in Rural Areas of South Mediterranean Countries: A Copula-Based GARCH Model

**DOI:** 10.3390/ijerph17165855

**Published:** 2020-08-12

**Authors:** Fabian Capitanio, Giorgia Rivieccio, Felice Adinolfi

**Affiliations:** 1Department of Veterinary Medical Sciences, University of Naples Federico II, 80137 Naples, Italy; 2Department of Managerial Studies and Quantitative Methods, University Parthenope Naples, 80133 Naples, Italy; giorgia.rivieccio@uniparthenope.it; 3Department of Veterinary Medical Sciences, University ALMA Mater Bologna, 40064 Ozzano dell’Emilia, Italy; felice.adinolfi@unibo.it

**Keywords:** food prices, price volatility, food riots, social unrest

## Abstract

Many discussions following the 2007/08 food price crisis have revolved around the magnitude of the negative impacts that it may have had on food security worldwide. In South-Eastern Mediterranean countries (SEMC), food security is strongly interrelated with several key economic and political issues. Many of these countries are becoming increasingly import-dependent, particularly on cereals, which are the essential raw material for human and animal food and feed. Due to both their economic system structure and consumption, the SEMC are responsible for a third of world cereals imports, whereas they account for only 5% of the world population. Given the set of constraints and this dependence on global markets, SEMC will be probably more exposed to severe swings in agricultural commodity prices in the coming years. In this view, this study examines the dependence structure among global food grain markets and Morocco and provides flexible models for dependency and the conditional volatility GARCH. A copula-based GARCH model has been carried out to estimate the marginal distributions of Morocco and world cereals commodity price changes. The results revealed that the joint co-movement between agricultural commodity price changes around the world and in Morocco, are generally considerable and there exists asymmetric tail dependence.

## 1. Introduction

The Food and Agriculture Organization of the United Nations (FAO) in 1996 [1] defined food security as situations where ‘‘all people at all times have the physical and economic access to sufficient, safe and nutritious food to meet their dietary needs and food preferences for an active and healthy life”. Thus, under food security, no individual faces hunger or starvation. Alternatively, food insecurity arises when some individuals face limited or uncertain access to nutritionally adequate and safe food. The FAO in 2015 reported that 795 million people were chronically undernourished in the years 2014–2016, representing 10.9% of the world population. Most live in developing countries. This is important since food security is an important component of the process of economic development [2] Over the last few decades, improvements in food security have come from an increase in agricultural productivity as well as a reduction in extreme poverty [3,4]. This scenario is worsened by several contingents and structural factors of today’s world. The last years have been characterized by a trend of increasing market instability for the agricultural market, illustrated by the commodities price volatility and the so-called food crisis in 2008. Volatility has a greater impact on developing countries and on the poor because it creates major import bill uncertainty.

Currently, around 20% of the world population use the opportunities of international trade to cover their demand for agricultural products.

In South-Eastern Mediterranean countries (SEMC), food security is strongly interrelated with several key economic and political issues. Many of these countries are becoming increasingly import-dependent, particularly on cereals, which are the essential raw material for human and animal food and feed. The SEMC are responsible for a third of world cereals imports, whereas they account for only 5% of the world population.

Given the set of constraints and this dependence on global markets, SEMC will be probably more exposed to severe swings in agricultural commodity prices in the coming years. In this view, this study examines the dependence structure among global food grain markets and Morocco and provides flexible models for dependency and the conditional volatility GARCH. Therefore, we implemented a copula-based GARCH model, which consists of estimating the marginal distributions of Morocco and world cereals commodity price changes, distinguishing the variety of cereal in durum and soft wheat and then estimating the copula parameters.

The issue of access to, and availability of, food on a global basis, known as food security, has been making headway on the political and media agenda since the start of the twenty-first century [5]. What gave cause for concern were the peaks in food prices experimented in 2007 and in 2010. After a long period of stagnating and declining agricultural prices, interrupted only by a spurt in the 1970s by the oil shock, since the mid-1990s an inverse trend started. The urgency to find solutions for feeding a projected population of more than 9 billion in 2050 became a global concern, gaining special focus by G7, G8, and G20 meetings [6]. The food price crises of 2007 and 2010 showed both the multiplicity and complexity of the underlying drivers. Some authors [7,8] have stressed the role played by structural factors in the medium and long term. Contributions in this area focused mainly on the relationship between the sustained growth in food demand expected for the coming years and the finite nature of natural resources available for production, sharpened by the growth of sustainability constraints [9,10]. This structural food gap is widely considered as the key factor in explaining the renewed interest in controlling natural resources recorded in the last ten years. On the other hand, there are also contingent factors playing a key role in shaping international food prices. The analysis of what happened in concomitance with the peaks recorded in 2007 and 2010 shows how those increases were the result of the action of several factors. Several authors have underlined the contribution given by the rise in the prices of energy products. Other studies have thoroughly investigated the role played by the commercial reaction, mainly export bans and import subsidies, in both food price rushes. Other scholars have been able to appreciate the impact on prices derived from the exchange rate trends. In both food crises, the role of climate events was also decisive. In Australia, the excessive rainfall and the frost experienced by Russia and Ukraine significantly reduced the global supply of cereals. Likewise, when in summer 2010, news spread of the great drought in Russia, Ukraine, and Kazakhstan, and the forecasts of the harvests were subsequently adjusted downward, the panic gripped the international food market [5]. Furthermore, the low rate of agricultural products traded in the global market fostered the impacts of the price increases. A large share of both maize and wheat produced in the world is traded on international markets and these volumes are concentrated in the hands of a few exporters. In this environment, price peaks could be fostered by market power and the return to the equilibrium may take a long time.

The Food and Agriculture Organization of the United Nations (FAO) defines food security as “a situation when all people, at all times, have physical, social, and economic access to sufficient, safe, and nutritious food to meet their dietary needs and food preferences for an active and healthy life” [1]. This definition implies evaluations both at a macro and micro level. At the macro level, food security is assured when food supplies are adequate to meet the food needs of the population. Domestic production and imports can contribute to this goal [11], in addition to possible international food aid. At the micro level, the theme of accessibility to food becomes crucial. Food availability is indeed a necessary but not sufficient condition [12] for food security. It must be accompanied by the access of a household to available food. Some authors [13] have noted how the impact of certain social factors, such as unemployment, low wages, and gender relations, reduce access to basic foods in North Africa Countries (NACs). Others [14] have shown that the effects of the price peaks of 2007 and 2010 have had negative effects on the quality of diets in important portions of the population of NACs.

The impact of price changes on the well-being of the most vulnerable groups of individuals depends on many factors [15], but particularly stressed has been the analysis of the relationship between the real increase in the price of a specific set of product and the share of income devoted to purchasing it [16]. The common result is that food prices can affect poverty and inequality through consumption and income channels [17,18,19]. Where socio-demographic dynamics have led to a concentration of the population in urban areas, the net position (between production and consumption of food) of most households has worsened [16,20]. As a result, the number of urban poor tends to increase in correspondence with increases in food prices [17,21]. For this reason, the food crises of 2007 and 2010 have had significant impacts on the state of global food insecurity [22]. The consequent impacts on the welfare of the population have been decisive in feeding food protests and riots in many areas of the African continent and in particular in North Africa Countries (NACs). The food riots were the opening of the Arab Spring. During the first pro-democracy rallies in December 2010, the Tunisian protesters brandished baguettes. A few weeks later a surge of 20–30 percent in the prices of staple foods in Algeria triggered the so-called ‘couscous revolt’, protesters in Jordan and Yemen shouted ‘bread and freedom’. This area is one of the most insecure regions in the world, mainly because of its dependence on food import [23], covering about 50% of the calorie consumption of its inhabitants. The dependence on imports makes the trade balance of NACs particularly sensitive to international food price fluctuations and the macroeconomic impacts associated with price increases can result in the sudden increase in the share of the population living below the poverty line. This area, together with the countries of the Middle East, is the largest importer of food in the world. The NACs average household food spending exceeds 30% of the overall expenditure and consequently, the increase in food prices can affect a significant portion of income. The region’s dependence on imports seems set to increase in the coming years and with it the vulnerability of NACs to food price shocks. In these countries, in fact, the opportunities to increase food self-sufficiency are limited by structural factors. These include the rapid increase in population and urbanization on the one hand and the lack of arable lands and water scarcity on the other. In addition, the food vulnerability of NACs has been exacerbated by their specific socio-economic conditions: the high rate of people under or close to the poverty line is the result of growing unemployment associated with preeminent social inequalities, then food price shock can be quickly translated into dramatic enlargements of the poverty area. This particular exposure to the risk of food insecurity was the main justification for the significant use of food subsidies in the NACs. Their massive use is Egypt, where in 2010, they exceeded 30% of total public spending and led the country to gain the appellative of “democracy of bread”. The definition indicates a model of a tacit social contract whereby in exchange for low cost or free essential services such as education, employment insurance, and health care, it is accepted by those it governs. The main symbol of this social contract was the system of subsidies that guaranteed bread and energy at an affordable price and allowed the Government to mitigate inflation. Direct and hidden costs of this type of social aid are strictly linked to the dynamics of international markets and to the level of inflation considered as socially acceptable. Prolonged food price increases and their impact on fiscal balances [24] could put at risk the viability of such instruments, and the inevitable decrease in subsides intensity could become socially unacceptable. The simultaneous growth in demand and prices in NACs that characterized the first decade of the 2000s has led to an increase in the fiscal cost of subsidies and has made it necessary to revise them. The fiscal inability of many NACs to continue to ensure social protection via food and energy subsidies has contributed to the breakdown of the social contract in which the Arab regimes had based a great part of their legitimacy [11,25].

From a statistical point of view, low correlations among price changes of food grains are historically observed but the dependence structure reveals itself in the form of the speed of shock transmission, not identifiable in a linear way. In this paper, we tried to evaluate the dependence structure among global food grain markets and Morocco; it would be relevant to know the dependence structure because it could be a risk management tool for speculators and a better forecasting tool for implementing public support for risk management and food security. The analysis focuses on the relationship between Morocco and world cereals commodity price changes, distinguishing the variety of cereal in durum and soft wheat.

We study the whole dependence structure of price change pairs by means of copula functions analyzing their extreme value co-movements; for this aim, we analyzed price data from the FAOSTAT database for crops from 1978 until 2018.

The Copula model allows for flexible, asymmetric, and then nonlinear dependencies making it possible to separate the joint dependence structure from its marginals and involve several families of univariate as well as joint distributions. In addition, copulas provide a measure of the financial shock contagion probability among economic markets by means of tail dependence measures. Morocco was selected basically because, since 1996, the Government liberalized cereals import from the world market, the cereals sector is one of the main sectors of agricultural production, especially in terms of employment in rural areas (30%), and Morocco is a net importer.

The structure of the paper is the following: Section 2 gives a description of the method of the research and a short overview of copulas and their main features. Section 3 shows the results of statistical analysis. Section 4 provides a discussion of the results, and then, some conclusions close the work in Section 5.

## 2. Materials and Methods

In order to investigate the world cereal import-dependence of SECM, we have analyzed weekly log-returns of durum and soft cereals in one of the SECM countries (Morocco) with respect to the same world commodity price changes. Morocco was selected basically because, since 1996, the Government liberalized cereals import from the world market, the cereals sector is one of the main sectors of agricultural production, especially in terms of employment in rural areas (30%), and Morocco is a net importer. Data cover the period from January 1998–July 2015, for 916 observations. Table 1 shows descriptive statistics.

To describe the shock transmission among global food grain markets and Morocco, we employ the copula functions which result in a very effective tool in modeling the dependence structure among financial markets due to the asymmetric dependence exhibited by food grain prices.

Copulas allow us to describe the whole dependence structure, also in the tails of the joint distribution, by-passing the classical assumptions imposed by traditional dependence models.

The great flexibility is due to the separate specification of the marginal behavior of univariate series from their joint distribution. This feature allows us to consider marginals of different forms within the same multivariate distribution, which can be nonlinear and asymmetric.

Their expression is defined by the following specification [26,27]:*H* (*x*_1, …,_*x_n_*) = *C* (*u*_1_, …, *u_n_*)
where *H* is a n-dimensional joint distribution function of the random variables *x_i_* (*i* = 1,…,*n*), *C* is the copula C of the uniform margins *u_i_* = *F_i_*(*xi*)~*U*(0,1) and *C* is unique if *F_i_* are continuous.

Several copula families were generally used to describe the joint dependence structure. Some of these families are elliptical and symmetrical, such as Gaussian and Student-t copulas, while others are not symmetrical with one parameter to measure the joint co-movement, such as the Archimedean copulas, Clayton, Gumbel and Frank, and some others, defined bi-parametric (BB), characterized by an Archimedean specification, such as the BB1 and BB7 copulas [28].

### 2.1. Copula Functions

#### 2.1.1. Elliptical Copulas

##### Gaussian Copula

The bivariate Gaussian Copula can be expressed in the form:$$C^{G}(u_{1},u_{2})=\Phi_{\rho x_{1}x_{2}}\left(\Phi^{-1}(u_{1}\right),\Phi^{-1}(u_{2}))$$
where $\Phi_{\rho x_{1}x_{2}}$ denotes the bivariate standard normal distribution applied to continuous random variables X_1_ and X_2_ with linear correlation $\rho_{x_{1}x_{2}}$, while $\Phi^{–1}$ is the inverse of the distribution function of each univariate standard normal distribution.

##### Student’s-t Copula

The copula of a bidimensional vector t-Student with υ degrees of freedom and $\rho_{x_{1}x_{2}}$ correlation is:$$C^{T}(u_{1},u_{2})=t_{\rho ,\nu }\left(t_{\nu }^{-1}\left(u_{1}\right),t_{\nu }^{-1}\left(u_{2}\right)\right)$$

#### 2.1.2. Archimedean Copulas

The Archimedean copulas family (Ling, 1965) can be built starting from the definition of a generator function Φ in R+, continuous, decreasing e convex, such that Φ(1) = 0.

The pseudo-inverse of Φ (Φ[−1]), which results continuous and decreasing in [0, +∞] and strictly decreasing in [0, Φ(0)], allows us to define the generic Archimedean copula function as:
*C^A^* (*u*_1_, *u*_2_) = [−1](Φ(*u*_1_) + Φ(*u*_2_))

In the class of the Archimedean copulas, it is possible to distinguish different copula functions, some of these with one parameter, the Gumbel, the Clayton and Frank copulas, some others with two parameters—denoted as BB copulas—such as Joe Clayton and Mixed Gumbel-Clayton

##### Gumbel Copula

The Gumbel copula was introduced by Gumbel in 1960 and belongs to Gumbel–Hougaard class.

It can be represented in the following way:$$C^{GU}(u_{1},u_{2})=exp-{\left[{\left(-ln\;u_{1}\right)}^{\alpha }+{\left(-ln\;u_{2}\right)}^{\alpha }\right]}^{1/\alpha }$$

The generator Φ is given by (−ln t)^α^ and the range of α is [1, +∞).

##### Clayton Copula

The expression of the Clayton copula can be written as:$$C^{C}(u_{1},u_{2})=max[\left(u_{1}^{-\alpha }+u_{2}^{-\alpha }-1{)}^{-1/\alpha },\;0\right]$$

The generator Φ(t) is represented by ($t^{-\alpha }-1)$ and Φ^−1^(t) = ($1+t{)}^{1/\alpha }$. If α > 0 it is completely monotonic.

##### Frank Copula

The Frank copula is different from the other copula functions, because the range of parameter value α is: (−∞,0) ∪(0,+∞). Its expression is given by:$$C^{F}(u_{1},u_{2})=-\frac{1}{\alpha }ln\left\{1+\left[\frac{\left(exp\left(-\alpha u_{1}\right)-1\right)\left(exp\left(-\alpha u_{2}\right)-1\right)}{\left(exp\left(-\alpha \right)-1\right)}\right]\right\}$$
where Φ(t) = $ln\frac{exp\left(-\alpha t\right)-1}{exp\left(-\alpha \right)-1}$.

#### 2.1.3. BB Copulas

The general formulation of BB copulas is:$$C^{BB}\left(u_{1},\;u_{2}\right)=\psi [-logK(exp\left(-\psi^{-1}\left(u_{1}\right),exp\left(-\psi^{-1}\left(u_{2}\right)\right)\right]$$
where K is max-infinitely divisible and ψ belongs to the class of Laplace Transforms.

##### BB1 Copula

Let K be the Clayton family and ψ be Laplace Transforms C (Joe, 97, p.375), then the Joe Clayton copula is: $$C^{BB1}\left(u_{1},\;u_{2}\right)={\left[1+{\left({u_{1}}^{-\alpha }+{u_{2}}^{-\alpha }-1\right)}^{1/k}\right]}^{-1/\alpha }$$
where α > 0, k ≥ 1. For k = 1, we get the popular Clayton copula. The generator function and its inverse are, respectively, $\Phi \left(t\right)={\left(t^{-\alpha }-1\right)}^{k}$, $\Phi^{-1}\left(t\right)={\left(1+t^{1/k}\right)}^{-1/\alpha }$ and α > 0, β ≥ 1.

##### BB6 Copula

$$C^{BB6}\left(u_{1},\;u_{2}\right)=1-{\left\{1-exp\left\{-[{\left(-log(1-(1-u_{1}\right)}^{\alpha }){)}^{k}+{\left(-log(1-(1-u_{2}\right)}^{\alpha }){)}^{k}{]}^{1/k}\right\}\right\}}^{1/\alpha }$$

where α≥1,β≥1.

##### BB7 Copula

Let K be the Gumbel family and ψbe Laplace Transforms B (Joe, 97, p.375), then:$$C^{BB7}\left(u_{1},\;u_{2}\right)=1-{\left\{1-{\left\{{\left[1-{\left(1-u_{1}\right)}^{\alpha }\right]}^{-k}+{\left[1-{\left(1-u_{2}\right)}^{\alpha }\right]}^{-k}-1\right\}}^{-1/k}\right\}}^{1/\alpha }$$
where $\Phi \left(t\right)={\left[1-{\left(1-t\right)}^{\alpha }\right]}^{-k}-1$ and Φ ^−1^(t) = 1− [ 1−(1 + t)^−1/k^ ]^1/a^ and α≥0,β≥1.

##### BB8 Copula

$$C^{BB8}\left(u_{1},\;u_{2}\right)=\frac{1}{\beta }\left\{1-{\left\{1-{\left[1-{\left(1-\beta \right)}^{\alpha }\right]}^{-1}[1-{\left(1-\beta u_{1}\right)}^{\alpha }][1-{\left(1-\beta u_{2}\right)}^{\alpha }]\right\}}^{1/\alpha }\right\}$$

where α≥1, 0≤β≤1.

### 2.2. Tail Dependence of Copulas 

Tail dependence is defined as a measure of concordance between less probable values of variables, which tends to concentrate in the lower and upper quadrant tails of the joint distribution.

Some copulas have null tail dependence such as Gaussian and Frank copulas, while others can have symmetrical and asymmetrical tail dependence coefficients. The Student’s t copula has equal coefficients, while Clayton shows only lower tail dependence and Gumbel only upper tail dependence. Finally, BB copulas have both nonnull tail dependencies with different coefficients for the upper and lower tails (to see the coefficient values see [28,29,30].

In a bivariate context, let Fi(·) (i = 1,2) be the marginal distribution function of two random variables X_1_ and X_2_ and let u be a threshold value, then the upper tail coefficient is [28]:$$\lambda_{U}=\underset{u\to 1^{-}}{lim}P(F_{1}\left(X_{1}\right)>u|F_{2}\left(X_{2}\right)>u)=\underset{u\to 1^{-}}{lim}P(U_{1}>u|U_{2}>u)$$
when λ_U_ is in (0,1], X_1_ and X_2_ are asymptotically dependent on the upper tail; if λ_U_ is null, X_1_ and X_2_ are asymptotically independent. 

Hence:$$P\left(U_{1}>u|U_{2}>u\right)=\frac{P(U_{1}>u,U_{2}>u)\;}{P(U_{2}>u)}=\frac{1-P\left(U_{1}\leq u\right)-P\left(U_{2}\leq u\right)+P\left(U_{1}\leq u,U_{2}\leq u\right)\;}{1-P\left(U_{2}\leq u\right)}$$

In terms of copula and its survival (Ĉ), we get:$$\lambda_{U}=\underset{u\to 1^{-}}{lim}\frac{\hat{C}\left(1-u,1-u\right)}{1-u}=\underset{u\to 1^{-}}{lim}\frac{1-2u+C\left(u,u\right)}{1-u}$$

The concept of lower tail dependence can be defined, in a similar way, as:$$\lambda_{L}=\underset{u\to 0^{+}}{lim}P(F_{1}\left(X_{1}\right)\leq u|F_{2}\left(X_{2}\right)\leq u)=\underset{u\to 0^{+}}{lim}P(U_{1}\leq u|U_{2}\leq u)$$

Therefore: $$P\left(U_{1}\leq u|U_{2}\leq u\right)=\frac{P\left(U_{1}\leq u,U_{2}\leq u\right)}{P\left(U_{2}\leq u\right)}$$

Then, in terms of the copula, we have:$$\lambda_{L}=\underset{u\to 0^{+}}{lim}\frac{C\left(u,u\right)}{u}$$

## 3. Results

Data are not stationary in mean because they show a positive trend (Figure 1). To avoid issues related to model serial dependence which assumes non-infinite variance in the parameter estimation procedure, data have to be transformed into detrended data. A way to delete the trend consists of applying first differences in case of non-stationarity of order one, typically observed in the presence of a trend.

Therefore, in the results of the Augmented Dickey-Fuller Test (see Table 2), there exists a non-stationary behavior of all time-series which can be removed by applying specific lag order differences. The analysis has to be conducted in terms, at least, of log price changes, as a measure of inflation, which is the log-difference between the price at time t and the price at time t − 1 (lag order = 1).

From Table 2, all time-series show positive mean returns while the skewness is positive only for durum cereal price changes. The kurtosis, higher than 3, which is considered as the minimum threshold for normal distribution, is relevant for all time-series, with the exception of world durum cereal. Soft cereals have very high kurtosis (leptokurtic distribution), whereas durum has a low of balanced kurtosis (mesocurtik distribution). This would imply a non-normal and asymmetrical behavior of time-series that has to be treated with non-gaussian distributions.

In the first step, we employ a GARCH type model to describe the marginal behavior of each return, modeling the conditional mean and variance and, then, we apply a copula function to join the margins into a multivariate distribution. Finally, we give a measure of dependence between extreme returns. The Copula-GARCH approach allows us to capture time-varying volatility and, eventually, leverage effect, and to fit the marginal distributions involved in the copula function (see, for the extended explanation of the model [30,31].

We specify the conditional means and variances for log-difference of the prices using an autoregressive and GARCH framework [31]. Table 3 reports main information criteria (AIC = Akaike Information Criterion, BIC = Bayesian Information Criterion, SIC = Schwartz Information Criterion, HIQ = Hannan Quinn Information Criterion) to select the best fitting model for innovations, excluding the gaussian innovation distributions, widely outperformed by the Skew-t and Student’s t distributions in all cases.

We find a GARCH (1,1) structure with Skew-t innovations (for which the minimum information criterion values is observed, Table 3) for world durum cereal log price changes (returns), a GARCH(1,1) with Student’s t innovations for world soft cereal returns, an ARMA(1,1)-GARCH(1,1) with Student’s t innovations for Morocco durum cereal returns and, finally, an ARMA(1,1)-GARCH(1,2) with Student’s t innovations for Morocco soft cereal log price changes.

A general expression of ARMA(m,d)-GARCH (*p*,q) model is the following:$$Y_{t}=\mu_{t}+\sigma_{t}\eta_{t}$$
$$\mu_{t}=\overset{m}{\underset{i=1}{\sum }}\phi_{i}Y_{t-i}+\overset{d}{\underset{j=1}{\sum }}\theta_{j}a_{t-j}$$
$$\sigma_{t}^{2}=\alpha_{0}+\overset{p}{\underset{i=1}{\sum }}\alpha_{i}{\left(Y_{t-i}-\mu \right)}^{2}+\overset{q}{\underset{j=1}{\sum }}\beta_{i}\sigma_{t-j}^{2}$$

Where innovations η_t_ are Skew-t and Student’s t distributed.

Table 4 shows the parameter estimates for the selected ARMA-GARCH models which result in all significant (*p*-values < 0.05). In Table 4, Shape and Skewness estimates are referred to as the distribution parameters of the ARMA-GARCH innovations (the skewness refers only to the Skew-t distribution).

We, then, select the residuals of the ARMA-GARCH models to join by a copula function in order to assess the whole dependence structure of returns.

The Kendall’s tau correlation coefficients (see Table 5) are significant for each pair of ARMA-GARCH standardized residuals, at the same lag (t) and at different lags (t vs. t − 1). This would imply a contemporaneous as well as a serial dependence of filtered Morocco cereal price changes from the world market which can be also nonlinear, whereas Pearson’s linear correlation coefficients (in Table 6) are almost always not significant (except for the contemporaneous dependence between Morocco and word durum grain).

Given these empirical relationships, we can model the dependence structure by copulas, useful extension and generalizations of conventional approaches for modeling asymmetric transmissions processes on the degree of market integration and its response to price shocks under the extreme market conditions [29,30,32,33,34].

To provide such a response to the impulse given by price shocks, a measure of concordance in the tails of the joint distribution, denoted as tail dependence, was derived. Since each copula has different behavior in the tails, providing several forms of tail dependencies, we have estimated all possible bivariate copulas, testing also the independence copula between return pairs at each time lag (t vs. t − i, for i = 0, 1, …10):

Zero tail dependence copulas, such as Gaussian and Frank, 90°/270°rotated Clayton, Gumbel, Joe, BB1, BB6, BB7 and BB8 were estimated, and, in particular:-Symmetric tail dependence copulas: Student’st-Asymmetric tail dependence copulas with:
positive tail dependence: Gumbel, Joe, BB6, BB8 and Survival Claytonnegative tail dependence: Clayton and Survival Gumbel, Joe, BB6 and BB8both tail dependencies: BB1, BB7, and their survivals

We present the results only for the copulas with the best fit to the data, according to the minimum AIC value.

Table 7 shows all significant parameter estimates (all *p*-values are < 0.05). Only the Survival Clayton shows an upper tail dependence coefficient different from zero and equal to 0.000245. This result reveals a weak positive tail dependence of Morocco durum cereal price changes in one week from those of the previous week in the world. Extreme changes (both positive and negative) of commodity prices in the world regarding soft cereals, do not have any impact on the Morocco prices, at the same time t as well as at different time lags.

## 4. Discussion

The results revealed that the joint co-movement between agricultural commodity price changes around the world and in Morocco, considering soft and durum cereal returns, are generally considerable and there exists an asymmetric tail dependence, especially for the durum selection. However, its tail dependence is relatively weak. Our findings have important implications for policy-makers, especially in geographical areas such as SEMC, which could be used to implement a better policy to optimize and stabilize the markets and hedge the welfare of people, especially in rural areas.

High price change volatility in all markets and a weak asymmetry.Strong and significant dependence among world food grain markets and Morocco.For durum wheat, a positive shock at time t (week) in the world can have an impact on the Morocco market at time t + 1 (next week).Improvements in the accuracy of price forecasts could be possible from considering alternative copulas to the Gaussian one.Need to build national policy aimed at tackling shock transmission (e.g., storage capability, futures market, irrigation)High dependence among markets could help as a risk management tool in future policy formulation and in price forecasting for both speculators in the commodity futures market and for policy-makers in the food importing market. This section may be divided by subheadings. It should provide a concise and precise description of the experimental results, their interpretation, as well as the experimental conclusions that can be drawn.

## 5. Conclusions

In the SEMC, food security is strongly interrelated with several key economic and political issues. Many of these countries are becoming increasingly import-dependent, particularly on cereals, which are the essential raw material for human and animal food and feed. The SEMC are responsible for a third of world cereals imports, whereas they account for only 5% of the world population.

Given the set of constraints and this dependence on global markets, SEMC will be probably more exposed to severe swings in agricultural commodity prices in the coming years. In this view, this study has examined the dependence structure among global food grain markets and Morocco, as well it provides flexible models for dependency and the conditional volatility GARCH. For this purpose, we implemented a copula-based GARCH model, which consists of estimating the marginal distributions of Morocco and world cereals commodity price changes, distinguishing the variety of the cereal in durum and soft wheat and then estimating the copula parameters. The main results achieved carrying out our empirical model revealed that the joint co-movement between agricultural commodity price changes around the world and in Morocco, considering soft and durum cereal returns, are generally considerable and there exists asymmetric tail dependence especially for the durum selection. Our findings have important implications for policy-makers, especially in geographical areas such as SEMC, which could be used to implement a better policy to optimize and stabilize the markets and hedge the welfare of people, especially in rural areas.

Governments of MPCs use many policy instruments in order to mitigate the effects on consumers arising from fluctuations in global agricultural commodity prices. Those measures helped MPCs in isolating households from price volatility and food inflation. Currently, different policy interventions (especially extensive use of price subsidies, as well as measures aimed at managing and regulating food consumption, production and trade) are used by MPCs in this field.

The latest peaks experienced in international prices have consequently complicated the macroeconomic scenario, leading towards the extensive use of resources devoted to price subsidy measures and other instruments, including production subsidies, import protection cuts, and build-up of food reserves, taking away fiscal resources that can be used to finance growth-enhancing investments [35]. The effects of food price transmission are linked to both the level and the typology of policy instruments used to mitigate the transmission into the domestic market. In the case of prolonged periods of increasing food prices, the amount of resources for covering the cost of those measures increases, generating a heavy fiscal drain on the government budget and compromising the sustainability of these kinds of responses.

Inevitably, fiscal and inflationary pressure has grown in many MPCs that are experiencing a fast-growing domestic food demand and spending a relevant share of their GDP on food subsidies. Some MPCs with high food import dependence and large fiscal deficits, such as Libya, Jordan, Lebanon, Egypt, Algeria, and Tunisia, appear most vulnerable to a sustained food price shock [36,37].

In this scenario, non-linearities and asymmetric adjustment remain an important issue to be explored especially when the objective of the research is to provide a price transmission mechanism that can be incorporated in a structural partial equilibrium model. Although asymmetric adjustment may also be the outcome of market imperfections, it is plausible that price support policies result in positive and negative changes in the international price affecting the domestic market in different ways. More importantly, such policies may imply a “threshold” or a minimum price, above which, the transmission of price signals takes place. Such a discrete adjustment process implies that movements towards a long-run equilibrium do not take place at all points in time, but only when the divergence from equilibrium exceeds a certain threshold. Policies such as price support mechanisms and tariff-rate quotas may result in such an adjustment process. In the former case, governments may intervene in the market when market prices fall below a floor level, whilst in the latter, international price signals pass-through when import volumes are sufficiently within or out of the quota. Thus, future research may focus on two-regime threshold cointegration, which may be beneficial, as it provides additional information in the form of the threshold if the objective of the analysis is the development of price transmission mechanisms for structural models.

## Figures and Tables

**Figure 1 ijerph-17-05855-f001:**
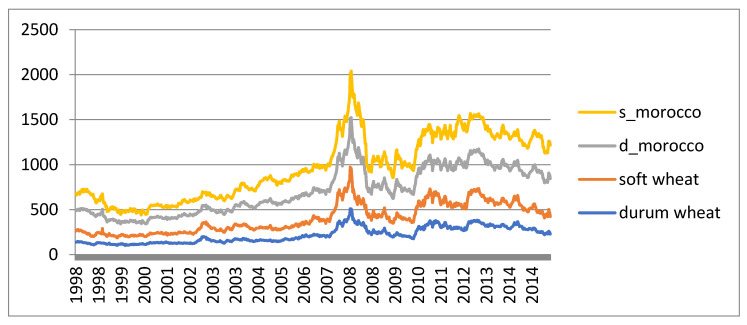
Morocco and world cereal prices.

**Table 1 ijerph-17-05855-t001:** Descriptive results.

Return	Mean	St.Dev	Skewness	Kurtosis
World_durum wheat	0.0006	0.035	0.060	1.290
World_soft wheat	0.0005	0.0483	−0.07	14.63
Morocco_durum wheat	0.0007	0.026	0.080	3.540
Morocco_soft wheat	0.0008	0.0336	−0.42	12.84

**Table 2 ijerph-17-05855-t002:** Augmented Dickey-Fuller test.

Price	ADF Test	*p*-Value
durum wheat world	−2.6584	0.2996
soft wheat world	−3.0649	0.1275
durum wheat Morocco	−3.1945	0.08888
soft wheat Morocco	−3.1551	0.09566

**Table 3 ijerph-17-05855-t003:** ARMA GARCH model comparison.

	Innovation Distribution	AIC	BIC	SIC	HQIC
durum wheat world	Skew-t	−3.989	−3.962	−3.989	−3.979
	Student′s-t	−3.987	−3.966	−3.987	−3.979
soft wheat world	Skew-t	−3.460	−3.433	−3.460	−3.450
	Student′s-t	−3.460	−3.439	−3.460	−3.452
durum wheat Morocco	Skew-t	−4.895	−4.858	−4.895	−4.881
	Student’s-t	−4.897	−4.866	−4.898	−4.885
soft wheat Morocco	Skew-t	−4.501	−4.459	−4.501	−4.485
	Student′s-t	−4.503	−4.466	−4.503	−4.489

**Table 4 ijerph-17-05855-t004:** ARMA-GARCH parameter estimates.

	World_Durum Wheat	World_Soft Wheat	Morocco_Durum Wheat	Morocco_Soft Wheat
α_0_	0.00 *	0.00 **	0.00 *	0.00 *
	(0.00)	(0.00)	(0.00)	(0.00)
α_1_	0.13 ***	0.15 ***	0.16 ***	0.25 ***
	(0.04)	(0.04)	(0.04)	(0.06)
β_1_	0.82 ***	0.73 ***	0.85 ***	0.28 *
	(0.05)	(0.07)	(0.03)	(0.14)
β_2_				0.49 ***
				(0.12)
Skweness	1.09 ***			
	(0.05)			
Shape	10.00 **	6.86 ***	4.39 ***	4.00 ***
	(3.13)	(1.30)	(0.73)	(0.60)
φ_1_			0.54 ***	0.52 ***
			(0.13)	(0.14)
θ_1_			−0.34 *	−0.33 *
			(0.15)	(0.16)
Num. obs.	915	915	915	915
AIC	−3.99	−3.46	−4.90	−4.50
Log Likelihood	−1831.73	−1588.57	−2245.78	−2066.18

*** *p* < 0.001, ** *p* < 0.01, * *p* < 0.05–St.errors in parentheses.

**Table 5 ijerph-17-05855-t005:** Kendall’s tau coefficients.

	Durum Wheat		Soft Wheat	
	Estimate	*p*-Value	Estimate	*p*-Value
World_t_ vs. Morocco_t_	0.051 *	0.020	0.047 *	0.032
World_t−1_ vs. Morocco_t_	0.043 *	0.049	0.046 *	0.037

* *p* < 0.05.

**Table 6 ijerph-17-05855-t006:** Pearson’s correlation coefficients.

	Durum Wheat		Soft Wheat	
	Estimate	*p*-Value	Estimate	*p*-Value
World_t_ vs. Morocco_t_	0.091 **	0.006	0.047	0.159
World_t−1_ vs. Morocco_t_	0.054	0.106	0.058	0.081

** *p* < 0.01.

**Table 7 ijerph-17-05855-t007:** Copula parameters and standard errors.

	Durum Wheat		Soft Wheat	
	Gaussian Copula		Frank Copula	
Return pair	Estimate	St.error	Estimate	St.error
World_t_ vs. Morocco_t_	0.093 **	0.033	0.430 *	0.198
World_t−1_ vs. Morocco_t_	Survival Clayton (180° rotated Clayton)		Frank copula	
	0.083 *	0.039	0.419 *	0.200

** *p* < 0.01, * *p* < 0.05.

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
