# Peer review of "Food Price Volatility and Asymmetries in Rural Areas of South Mediterranean Countries: A Copula-Based GARCH Model"

_ijerph, 2020, doi:10.3390/ijerph17165855_

Round 1

Reviewer 1 Report

Brief Summary

this study examines the dependence structure among global and Moroccan food grain markets thorughout bivariate copula functions of ARMA-GARCH univariate models. Results suggest some degree of rank correlation among univariate time series of log price changes at same lag and at one lag. Furthermore, results suggest an ARMA-GARCH structure in univariate time series of log price changes, and a weak positive tail dependence of Moroccan durum cereal price changes in one week from lagged World durum cereal price changes.

Broad comments

While the underlying research seems to match high scientific standards, the quality of presentation should be consistently improved, as it does not allow to appreciate fully the empirical evidence built (that seems to have, instead, high relevance and scientific soundness).

First of all, political framing is "fragile" and many issues are questionable, as those one concerning the acceptance of authoritarian regimes in exchange for the provision of basic welfare services as food and energy (Lines 123-133). The international political debate criticized this model and pushed for more democratic and development-oriented institutions in the MENA (NAWA) region. Authors are kindly asked to improve the analysis of the geopolitical scenario.

Second, most econometric methods used in the empirical analysis are non-standard to a reader that is not familiar with advanced methods in econometric. Therefore, Authors are kindly asked to provide extensive explanation of methods, results, and of their interpretation. Specific attention should be paid in adding equations within text and p-values in tables (of course, the latter can be easily computed, but the unexperienced reader should not read the paper while computing tests and consulting statistical tables to verify their significance).

Third, it seems that comovements are identified trhougout Kendall's tau rather than using copula based analysis of ARMA-GARCH models. Authors are kindly invited to provide further explanation on the relevance of ARMA-GARCH analysis and of copula based models of tail dependence in times of macroeconomic instability (indeed, Authors already explain this issue, but more "weight" should be given to this point, otherwise a reader interested only in non-technical issues might skip the empirical part and go straight to conclusions).

In sum, to appreciate the scientific soundness of the underlying research, I suggest that Authors should add a borad illustration of theoretical econometric issues in Material & Methods and Results sections, otherwise the paper will be of interest only to a limited number of readers. Second, they should separate descriptive analysis from empirical analysis, highlighting in the interpretation of descriptive results the motivation to go further with the analysis of tail dependence. Finally, they should improve the connection among the final statements and the interpretation of the empirical results provided.

Specific comments

Abstract is repeated in introduction (see from Line 33) and should be at least rephrased.

More in general the structure of "introduction" should be improved (for example, if the text quoted in the abstract is deleted, Line 47 might be placed at the beginning).

Line 73-76: this sentece is not clear and should be rephrased: if "only" 12% of maize and 18% of wheat are traded internationally there should not be relevant shocks on domestic prices. May be, instead that 12% and 18% are large quotas.

Line 182-183: ADF is a test for non-stationarity. Few sentences might help the unexperienced reader to clarify the relevance of stationarity and the need to consider first differences in case of non stationarity of order one. Furthermore, results of the ADF might be added in a table to illustrate how several lags (up to order n) has been considered.

Line 187: Table 1 deserves further interpretation of results: log price changes are a measure of inflation, soft cereals have very high kurtosis (leptokurtic distribution), whereas durum have low of balanced kurtosis (platikurtic/mesocurtik distribution)

Line 192: Also Table 2 deserve further explanation of Kendall's tau and interpretation of results. This seems to be a key point, as comovements are identified.

At the end of descriptive analysis involving non-stationarity and comovements, few sentences might be added to motivate (or to remind the motivation for) further analysis on tail dependence. Moreover, the copula based ARMA-GARCH approach should be better illustrated, as well as the decision to opt for this approach (are there other alternatives available?). 

Line 206: Equations of ARMA-GARCH models, as well as a general explanation (especially on non-nonormal innovations) should be added before the table to allow the unexperienced reader to better interpret results. Moreover, within the table p-values should be added.

Lines 1-11 p.7: Authors should introduce copula function and provide equations for the bivariate copulas tested.

Line 12 p.7: Table 4 might be integrated with an additional table in Appendix illustrating at least a subset of less significative estimates.

Lines 15-19 p.7: this is a key issue and deserves further comments.

Lines 27-37 p.7: Findings should be put in a relationship with empirical results. Of course, the reader can guess the association, but the latter is not straightforward.

I have noticed many typos within the text, Authors are kindly invited to improve editing.

Line 10: there is a typo (delete "A single")

Line 33-34: this sentence is decontextualized and it may be removed from text.

Line 46: "estimated" instead of "estimates"

Line 72: please modify "has exacerbates"

Line 108: may be "calories"?

Line 126: Please rephrase "a highly power is accepted by those it governs".

Line 133: "led "instead of "lead"

Line 151: "very relevant tool" (may be effective tool?)

Line 37-39 (after break): delete text from "this section" to "drawn".

Line 93-94 should be filled in

Reviewer 2 Report

The paper “Food Price Volatility and Welfare Implications in Rural Area in South Mediterranean Countries” aims to examine the dependence structure among global food grain markets and Morocco and to provide flexible models for dependency and the conditional volatility GARCH. In its present form the paper shows different shortcomings.

The introduction is very confusing. Authors are invited to better specify the context of their research and the aims. Then a literature review should be provided. As it is now everything is mixed up and the reader gets lost.

Methodology section: no information about the data that has been used

Discussion: very poor. Authors are invited to better highlight their results and to provide a clear evidence of the advancements they made.

Reviewer 3 Report

This is an interesting paper. The interpretation of the results could be improved by providing more detail linking the empirical methods to the results. Motivate the importance of the approach in deciphering results.

Some editing is needed (e.g., first sentence of the abstract).

This paper provides estimates of the dependence in cereal markets returns (price changes) in Morocco and the rest of the world. Results indicate significant joint dependence between Morocco and world returns in durum and in soft cereals markets, with some (weak) positive tail dependence for durum cereals. This is an interesting paper that uses advanced econometric techniques to find important results in understanding co-movements in price
changes across Morocco and the rest of the world.

The title of the paper could more accurately represent the contents of the paper.

P. 5, line 179f, “…we have analysed weekly log-returns of durum and soft cereals in one of the SECM countries (Morocco) with respect to the same World commodity price changes. Data cover the period: Jan 1998-July 2015, for 916 observations.”

Table 1 provides descriptive statistics of returns for World durum and soft cereals, and Moroccodurum and soft cereals. However, no source is provided for the data. The source of the data should be provided in the paper. Also, the description of the data should not be in the results section, it should be earlier in the paper.

The interpretation of the results could be improved by providing more detail linking the empirical methods to the results. Continuing in the results, section 3 of the paper, ARMA-GARCH parameter estimates are presented but the estimated models are never specified in the paper. The specified models used for these estimated results should be presented prior to the results, in the methods section of the paper (currently section 2), with the copulas development.

Motivate the importance of the econometric approach in
deciphering unique results in the paper.

Explain why these results are important in forecasting returns (price changes) in these markets.

There are numerous typos and omissions that need to be corrected, and further editing is needed, e.g.,
1. the first sentence of the abstract, “A single Currently 16% of the world
population…” doesn’t make sense.
2. Abstract, line 16, “SEMC will be probably more exposed…” more appropriately
would be “SEMC will probably be more exposed…”
3. P. 9, lines 83 and following, author contributions are not provided.
Improving the general presentation and description of what appears to be quality research would help reach a wider audience or readers of the journal.

My comments above are intended to be constructive in this regard.

Round 2

Reviewer 2 Report

Thank you for submitting this new version of the manuscript and for addressing all the suggestions provided.